# Comparative Evaluation of *Candida* Species-Specific T-Cell Immune Response in Human Peripheral Blood Mononuclear Cells

**DOI:** 10.3390/biomedicines12071487

**Published:** 2024-07-05

**Authors:** Balaji Pathakumari, Weida Liu, Qiong Wang, Xue Kong, Guanzhao Liang, Santosh Chokkakula, Vasundhara Pathakamuri, Venkatrao Nunna

**Affiliations:** 1Department of Medical Mycology, Institute of Dermatology, Chinese Academy of Medical Science and Peking Union Medical College, Nanjing 210042, China; liumyco@hotmail.com (W.L.); chongziqiong@163.com (Q.W.); kxcmxzx@163.com (X.K.); guanzhaoguan@126.com (G.L.); 2Division of Pulmonary and Critical Care Medicine, Department of Medicine, Mayo Clinic, Rochester, MN 55905, USA; 3Department of Microbiology, Chungbuk National University, College of Medicine and Medical Research Institute, Cheongju 28644, Republic of Korea; biochemsanthu@gmail.com; 4Department of Radio-Diagnosis, Sri Venkateshwara Medical College, Tirupathi 517507, India; vasundharapathakamuri3@gmail.com; 5Division of Nephrology, Department of Medicine, Washington University School of Medicine, St. Louis, MO 63110, USA; nunnabiotech@gmail.com

**Keywords:** invasive candidiasis, immunotherapeutics, non-albicans *Candida* species, T-cell immunity, ELISA, flow cytometry

## Abstract

Non-albicans *Candida* (NAC) species are increasingly recognized as significant contributors to candidemia infections; however, relatively less is known about the immune responses induced by these species. In this study, we compared the cytokine production ability of human peripheral blood mononuclear cells (PBMCs) upon stimulation with different *Candida* species (*Candida* spp.). We measured secreted cytokines using ELISA and checked the functional profiles of T-cell responses using multicolor flow cytometry. Although there was a differential expression of cytokines against *Candida* spp., significant difference were observed in the levels of IFN-γ, TNF-α, IL-10, IL-12p40, and IL-23 (*p* < 0.05) between *Candida* spp. A significant difference was observed between *C. albicans* and *C. glabrata* (*p* = 0.026) in the levels of TNF-α. *C. glabrata* showed significant differences compared to *C. albicans*, *C. parapsilosis*, and *C. krusei* in the levels of IL-10 (*p* values of 0.02, 0.04, and 0.01, respectively). Despite the percentages of CD4^+^ and CD8^+^ expressing Th1, Th2, and Th17 cytokines being higher in stimulated PBMCs, none of the *Candida* spp. showed significant differences. The levels of secreted IL-17A and IL-23 were consistently lower in *Candida* spp. regardless of the stimulus used. Here, we showed the differential regulation of Th1, Th2, and Th17 during *Candida* spp. stimulation of the immune system ex vivo. Additionally, our findings suggest that *C. albicans* elicits an IFN-γ response, whereas *C. glabrata* promotes IL-10 cellular responses, but this warrants additional studies to conclude this association. This investigation holds the potential to advance our comprehension of the distinct immune responses induced by *Candida* spp., with probable implications in designing antifungal immunotherapeutics.

## 1. Introduction

*Candida* species are the most common human fungal pathogens and cause both superficial (mucosal and cutaneous) and systemic infections [1]. Approximately 92% of candidemia infection cases are caused by *Candida albicans* (*C. albicans*), *C. glabrata*, *C. tropicalis*, *C. parapsilosis*, and *C. krusei* [2]. *Candida* spp. represent the most frequent cause of invasive fungal infections worldwide [3], responsible for three types of candidiasis, i.e., thrush, vaginal candidiasis, and invasive candidiasis (IC). Invasive *Candida* infection is the most common type of bloodstream infection among invasive fungal infections (IFIs) [4]. IC is a serious infection that causes life-threatening diseases and is responsible for the significant mortality rates (40%) in immunocompromised patients [5]. While invasive candidiasis infection is more commonly caused by *C. albicans*, half of the reported cases of candidemia are now attributed to non-albicans *Candida* (NAC) species [6], and the frequency of each species varies with geographic regions [7].

T lymphocytes are relevant for protection against fungal infections and the development of emerging vaccines focused on inducing strong and durable cellular immunity. CD4 T-cells are indispensable for host survival, while CD8 Tcells are necessary for immune defense against pathogens [8]. Activated T-cells secrete proinflammatory cytokines, which drive the differentiation of naïve T-cells into CD4 T helper (Th) subsets. The Th1 response, characterized by the production of proinflammatory cytokines such as IFN-γ, IL-2, and TNF-α, is associated with a protective immune response against *Candida* spp. [9]. The role of the Th2 immune response is more associated with an increased fungal burden and disease exacerbation [10]. Th2 cells mainly produce anti-inflammatory cytokines such as IL-4, IL-5, IL-10, TGF-β, and IL-13. The cytokines IL-4 and IL-13 drive the differentiation of naive T-cells toward the Th2 subset and inhibit Th1-type differentiation, resulting in reduced proinflammatory cytokine production. The discovery of the Th17 subset has had a significant impact on the understanding of fungal immunology, and this subset produces IL-17A, IL-17F, IL-21, and IL-22 cytokines [11]. Similarly, in a Rag1^−/−^ mouse model of adaptive immunity, Th17 cells are protective against oropharyngeal candidiasis (OPC) and correlated with the clearance of *Candida* [12]. Furthermore, it was demonstrated that IL-17 receptor knockout mice were more susceptible to systemic *Candida* infection as compared to wild-type mice [13]. However, the role of IL-17 in protection against and the pathogenesis of fungal infections remains unclear. Th1/Th17 subsets aid in the recruitment of neutrophils and other immune cells for the clearance of fungal infections [14]. Thus, the Th1/Th17 paradigm is important in understanding protective immune responses. Th22 cells produce the IL-22 cytokine, which plays a role in local protection in mucocutaneous fungal diseases [15] and also mediates antifungal resistance in human vulvovaginal candidiasis (VVC) and recurrent VVC [16].

Along with *Candida albicans*, NAC species such as *C. glabrata*, *C. parapsilosis*, *C. tropicalis*, and *C. krusei* are emerging causes of invasive candidiasis. *C. parapsilosis* stands out as a prominent NAC species that is frequently responsible for neonatal candidemia in children [7] and also contributes to compromised immunity in adult patients [17]. Notably, *C. parapsilosis* is commonly associated with elevated mortality rates in cases of invasive candidiasis among patients with hematologic malignancies [18,19]. *C. glabrata* is an opportunistic *Candida* species. In contrast to *C. parapsilosis*, the incidence of *C. glabrata* is higher in adults and elders and often more resistant to common antifungals [12]. *C. tropicalis* accounts for 10–15% of blood stream *Candida* isolates in tropical and subtropical regions [19], commonly associated with patients with neutropenia and malignancy [20]. *C. krusei* is an uncommon opportunistic *Candida* pathogen, predominantly found in immunocompromised patients, especially in patients with leukemia [21]. Recently, the Centers for Disease Control and Prevention (CDC) issued a clinical alert to healthcare facilities concerning the rapid international emergence of *C. auris*, which is spreading at an alarming rate and is considered an “urgent antimicrobial resistance threat”. Recent studies have reported that 90% of *C. auris* isolates are resistant to at least one class of antifungal agents, 30% are resistant to at least two different classes, and there are even pan-resistant isolates [22]. *C. albicans* and *C. glabrata* induce human whole-blood IL-12 secretion, which plays a key role in NK cell activation and triggers degranulation of secretory granules [23]. Compared to *C. glabrata*, *C*. *albicans* elicits a strong PMN response through the upregulation of activation markers, secretion of antimicrobial compounds, and rapid phagocytosis [24].

The *Candida* genus represents a highly polyphyletic group and comprises a wide range of phylogenetically unrelated anamorphic fungi, and these species do not share a single evolutionary origin. *Candida* spp. have evolved independent strategies to survive as opportunistic pathogens in host environments [25]. Therefore, the genes which are central for survival in the host for one species may be irrelevant in another [26]. However, the clinically relevant *Candida* spp. have been reexamined by applying molecular technologies to review nomenclature changes over the past years. A better knowledge of the evolutionary relationships of *Candida* spp. is vital to understand the nature of their virulence, clinical relevance, and, therefore, their host immune responses [27]. Since multiple phenotypic variations are found between the cell wall compositions, growth requirements, and virulence factors of *Candida* spp., these variations may be responsible for the distinct immune responses of each species [28]. Furthermore, there has been an increase in the frequency of antifungal drug resistance in *Candida* spp. that poses a global threat. Since *Candida* spp. are heterogeneous, studying host defense responses to NAC species might help in addressing the disparities between species [28]. Great efforts have been made to understand the host defense mechanisms against *C. albicans*, but little is known about the immune responses induced by other NAC species, even though these have been considered an emerging threat to immunosuppressed and immunocompetent patients in recent years [29]. Therefore, understanding the host defense mechanisms of each *Candida* spp. is crucial to understand host–pathogen interactions, and induced immune response would be helpful in designing immunotherapeutics strategies [9]. In this study, our primary focus was to compare cytokine production in human peripheral blood mononuclear cells (PBMCs) following stimulation with the most prevalent NAC species. We pursued two main objectives: firstly, to determine the cytokine profiles secreted in response to different *Candida* stimulations using the enzyme-linked immunosorbent assay (ELISA), and secondly, to assess the functional profile of T-cell responses at the single-cell level, employing multicolor flow cytometry.

## 2. Materials and Methods

### 2.1. Fungal Strains and Culture Conditions

In this study, we used *C. albicans* (ATCC SC5314), *C. glabrata* (ATCC 2001), *C. parapsilosis* (ATCC 22019), *C. tropicalis* (ATCC-MYA-750), and *C. krusei* (ATCC 6258) species for the functional characterization of T-cell response. These *Candida* spp. were grown on Sabouraud agar (1% peptone, 4% glucose, 2% agar) and YPD agar medium (1% yeast extract, 2% peptone, 2% agar) for 24–48 h at 25 °C to 30 °C. The single colony was inoculated into the broth medium and allowed to grow overnight at 25 °C with shaking at 150 rpm. Cells were washed twice with PBS and treated via heat shock at 90 °C for 30 min to inactivate the cells. Cells were counted with a hemocytometer and the concentration was adjusted for stimulation assays. Because of the variation in the growth kinetics of *Candida* spp., and also *C. albicans* and *C. tropicalis* form hyphae in cell culture, we used heat-killed yeast for stimulations [30,31].

### 2.2. Study Participants

For this study, 5 females and 3 males > 18 years old in a stable medical condition were enrolled as study participants. The median age was 32 years. These individuals were carefully selected to ensure they were free from notable infections or suspected symptoms. Exclusion criteria encompassed patients with HIV positivity, individuals with diabetes, pregnant women, cancer, or autoimmune diseases that might impact the immune system. Additionally, individuals with a history of previous *Candida* infections, those who had undergone antifungal treatment, or those currently receiving immunosuppressive therapy were also excluded from the study. These conditions were confirmed by checking their past medical records and by completing a questionnaire. Blood specimens were collected from all the recruited participants after obtaining their informed consent. The study was conducted in accordance with the Declaration of Helsinki and approved by the Institutional Review Board, Institute of Dermatology, Chinese Academy of Medical Sciences. The study methods were approved by the institutional ethics committee, Chinese Academy of Medical Sciences (record no: 417-2019-004).

### 2.3. Peripheral Blood Mononuclear Cell Isolation

Peripheral blood was collected in heparin-coated tubes from the enrolled study participants. PBMCs were isolated using the Ficoll density gradient separation protocol. The buffy coat was collected, washed twice in PBS, and cell viability was determined by trypan blue exclusion test. Finally, cells were counted by a hemocytometer.

### 2.4. Ex Vivo Stimulation of PBMCs with Candida *spp.*

The concentration of PBMCs was adjusted to 1 × 10^6^ cells/mL with Rosewell Park Memorial Institute (RPMI) 1640 medium (Sigma-Aldrich, St. Louis, MO, USA). The cells were stimulated with heat-killed *Candida* spp. at a final concentration of 1 × 10^6^ cells/mL in a round-bottom 48-well plate. Cells were cultured under similar conditions without any stimulation and were denoted as unstimulated (US). The culture plate was incubated for 18 h at 37 °C with 5% CO_2_. For flow cytometric assays, we used Brefeldin A (BD Biosciences, San Diego, CA, USA), which was added 12 h before the termination of the culture. We evaluated the multicolor flow cytometric data of *Candida*-specific CD4^+^ and CD8^+^ T-cell cytokine recall responses, such as IFN-γ, TNF-α, IL-4, IL-17A, and IL-22, after 16 h ex vivo stimulation.

### 2.5. Quantification of Cytokines in Culture Supernatants

The cell-free supernatants were harvested, and subsequently, quantification of cytokines was performed by sandwich ELISA (R&D systems, Abbingdon, UK). We measured IFN-γ, TNF-α, IL-1β, IL-6, IL-10, and IL-12p40 after 24 h and IL-17A and IL-23 after 6 days in the ex vivo cultures of PBMCs with *Candida* spp. Due to insufficient blood, we performed the ELISA in 6 healthy PBMC samples.

### 2.6. Immunostaining and Flow Cytometry

The cells were stained with T-cell-specific surface markers (CD3, CD4, and CD8) and incubated for 30 min at 4 °C. Following surface staining, cells were washed with PBS, fixed, and permeabilized with cytofix/cytoperm buffer (BD Biosciences, San Diego, CA, USA). After fixing, the cells were washed with Perm/Wash solution (BD Bioscience, San Diego, CA, USA), and then stained for intracellular Th1 and Th2 cytokines (anticytokine mAbs cocktail containing IFN-γ, TNF-α, IL-4, IL-10, IL-17A, and IL-22). Cells without staining served as the unstained control. Finally, cells were fixed in 0.5% paraformaldehyde that was added to the cells and acquired using an Attune NxT Acoustic Focusing cytometer with Attune NxT software 3.1.2 (ThermoFisher, Waltham, MA, USA). We acquired at least 100,000 cells in the lymphocyte gate. The percentage of phenotypic and functional markers was analyzed by FlowJo software (Tree Star Inc., Ashland, OR, USA, version 7.1.1). The boundaries were defined by setting fluorescence-minus-one (FMO) controls for all the antibodies. Compensation was performed from fluorochrome-conjugated antibodies coupled with compensation beads.

### 2.7. Statistical Analysis

GraphPad Prism 5 software (GraphPad Software 5.00, San Diego, CA, USA) was used for statistical analysis. Nonparametric Kruskal–Wallis ANOVA coupled with Dunn’s correction was used for multiple comparisons. The differences in various cytokine levels between two stimulations were assessed by Mann–Whitney U test. The statistical difference was considered significant when the *p* values were <0.05.

## 3. Results

### 3.1. Ex Vivo Cytokine Response to Candida *spp.*

We quantified a comprehensive panel of secreted cytokines (IFN-γ, TNF-α, IL-1β, IL-6, IL-12p40, IL-17A, IL-23, IL-10) in both unstimulated and *Candida*-stimulated PBMCs from healthy controls. The median levels of various cytokines are illustrated in Figure 1 and Figure 2. All the cytokines showed a statistically significant difference between unstimulated and stimulated PBMCs (*p* < 0.05) for all the *Candida* spp. Significant differences were observed in the levels of IFN-γ, TNF-α, IL-10, IL-12p40, and IL-23 (*p* < 0.05) between the different *Candida* spp., indicating a distinct cytokine expression profile associated with each species. However, no significant differences were observed for IL-1β, IL-6, and IL-17A between the stimulations.

### 3.2. Differential Cytokine Responses to Candida *spp.*

*Candida* spp.-specific stimulation significantly elevated IFN-γ in PBMCs compared with the unstimulated control. Specifically, the median levels of species-specific IFN-γ were higher for *C. albicans* (596 pg/mL), *C. glabrata* (537 pg/mL), and *C. parapsilosis* (541 pg/mL) than the samples treated with *C. tropicalis* (333 pg/mL) and *C. krusei* (221 pg/mL) (Figure 1a). Notably, a significant difference was observed in *C. albicans* and *C. glabrata* when compared to *C. krusei* (*p* = 0.008 and *p* = 0.01, respectively). Similar to IFN-γ, the other proinflammatory cytokine, TNF-α, also showed increased levels in *Candida*-stimulated samples (Figure 1b). Interestingly, the baseline TNF-α level is higher than that of any other cytokine measured in our study. The median TNF-α levels for *C. albicans*, *C. glabrata*, *C. parapsilosis*, *C. tropicalis*, and *C. krusei* were 2588 pg/mL, 1563 pg/mL, 2072 pg/mL, 1237 pg/mL, and 2083 pg/mL, respectively. However, a significant difference was observed between *C. albicans* and *C. glabrata* (*p* = 0.026), *C. albicans* and *C. tropicalis* (*p* = 0.0411), and *C. krusei* and *C. tropicalis* (*p* = 0.0411). The other proinflammatory cytokines such as IL-1β and IL-6 showed a significant difference between *Candida*-stimulated and unstimulated samples but not among *Candida* spp. (Figure 1c,d). The median levels of IL-12p40 were higher in samples stimulated by *C. albicans* (154 pg/mL), *C. glabrata* (117 pg/mL), and *C. parapsilosis* (210 pg/mL) than those stimulated by *C. tropicalis* (23 pg/mL) and *C. krusei* (17pg/mL). Thus, a significant difference was observed in *C. albicans* when compared with *C. tropicalis* (*p* = 0.002) and *C. krusei* (*p* = 0.002), and *C. glabrata* when compared to *C. tropicalis* (*p* = 0.01) and *C. krusei* (*p* = 0.004). Similarly, we found a significant difference in *C. parapsilosis* vs. *C. tropicalis* (*p* = 0.002), and *C. parapsilosis* vs. *C. krusei* (*p* = 0.002) (Figure 2b).

Remarkably, the baseline levels of anti-inflammatory cytokine IL-10 were notably lower than the proinflammatory cytokines. *C. glabrata* induced higher levels of IL-10 (369 pg/mL) secretion than any other *Candida* spp. *C. glabrata* showed significant differences compared to *C. albicans* (*p* = 0.02), *C. parapsilosis* (*p* = 0.04), and *C. krusei* (*p* = 0.01), whereas *C. tropicalis* displayed a significant difference compared to *C. krusei* (*p* = 0.04) (Figure 2a).

To investigate the role of the Th17 subset in immunity against *Candida* spp., we analyzed the production of IL-17A and IL-23 after stimulating PBMCs with *Candida* for 6 days. The levels of IL-17A and IL-23 were lower than other inflammatory and anti-inflammatory cytokines across all stimulations (Figure 2c,d). The median levels of IL-17A for *C. albicans*, *C. glabrata, C. parapsilosis*, *C. tropicalis,* and *C. krusei* were 111 pg/mL, 144 pg/mL, 134 pg/mL, 80 pg/mL, and 74 pg/mL, respectively. However, none of these *Candida* spp. showed a significant difference between the stimulations. Interestingly the cytokine IL-23 showed a significant difference for *C. glabrata* when compared with *C. parapsilosis* (*p* = 0.01) and *C. krusei* (*p* = 0.02) (Figure 2d).

### 3.3. Functional Signature of Candida-Specific CD4^+^ and CD8^+^ T-Cells

In this study, we also quantified the proportion of CD4^+^ and CD8^+^ cells secreting Th1, Th2, and Th17 cytokines in *Candida* spp. stimulated PBMCs among healthy controls. We compared the expression of intracellular cytokines produced by CD4^+^ and CD8^+^ T-cells across *Candida* spp. The frequency of *Candida*-specific CD4^+^ and CD8^+^ cells secreting cytokines was higher in stimulated PBMCs compared to the unstimulated cytokine response. The general gating strategy followed for selecting CD4^+^ and CD8^+^ T-cells is depicted in Figure 3.

The proportion of IFN-γ-producing CD4^+^ and CD8^+^ T-cells was elevated in *Candida*-stimulated PBMCs, while no such response was observed in the unstimulated control (Figure 4a,b). The median levels of CD4^+^ and CD8^+^ secreted IFN-γ are shown in Table 1. The median levels of CD4^+^ and CD8^+^ secreted IFN-γ were high in *C. albicans* compared to other *Candida* spp. though without statistical significance between the stimulated samples. TNF-α is a potent monocyte/macrophage activator that works synergistically with IFN-γ to induce antimicrobial activity. *Candida*-specific TNF-α was increased in stimulated samples as compared to the control group (*p* < 0.05). While CD4^+^ TNF-α producing T-cell levels were elevated in all stimulations compared to CD8^+^, none of the stimulations exhibited statistical significance (Figure 4c,d). In contrast to IFN-γ, the basic levels of CD4^+^ and CD8^+^ TNF-α producing Tcells were higher in both stimulated and unstimulated PBMC samples. Though the percentage of CD4^+^ and CD8^+^ IL-12^+^-secreting T-cells was higher in stimulated PBMCs, no significant difference was observed between *Candida* spp. in any of the stimulations.

In another set of analyses, we compared the frequency of *Candida* spp.-specific CD4^+^ cells secreting IL-4 and IL-12. The median levels of CD4^+^ and CD8^+^ IL-4^+^-secreting T-cells were elevated in *Candida*-stimulated compared to unstimulated samples. However, due to minimal differences observed among the stimulations, no statistically significant variations were found (Figure 4e,f). It is worth noting that baseline levels of CD4^+^ and CD8^+^ IL-4-secreting cells were notably high even in unstimulated samples (Figure 4e,f). As compared to CD4^+^, a higher frequency of IL-4-secreting CD8^+^ T-cells was observed in all stimulations. This suggests that CD8^+^ T-cells could be the major cell population secreting IL-4. The percentages of CD4^+^ and CD8^+^ IL-17A^+^ levels were high in stimulated PBMCs compared to unstimulated samples. However, no significant differences were observed between *Candida* spp. in any of the stimulations. Intracellular staining for CD4^+^ and CD8^+^ IL-22^+^ revealed a lower number of IL-22 producing cells in the CD4^+^ and CD8^+^ T-cell population after treatment with *Candida* spp.

## 4. Discussion

Studying the immune responses to invasive candidiasis is an ongoing and exciting challenge in the design of antifungal immunotherapeutics strategies. Due to the increase in incidence, morbidity, and mortality associated with NAC species, unraveling similarities and differential immune responses to these species may help to identify novel targets. In this study, we investigated and compared the capacity of *Candida* spp. to stimulate cytokine production in human PBMCs. We specifically examined the production of both proinflammatory and anti-inflammatory cytokines, including IFN-γ, TNF-α, IL-1β, IL-4, IL-8, IL-10, IL-12p70, IL-17A, and IL-23. Further, we evaluated the multicolor flow cytometric analyses of *Candida* spp. specific for CD4^+^ and CD8^+^ T-cell cytokine responses, such as IFN-γ, TNF-α, IL-4, IL-17A, and IL-22.

We examined the production of various inflammatory and anti-inflammatory cytokines after stimulating PBMCs with different *Candida* spp. Th1 response is characterized by the production of proinflammatory cytokines IFN-γ and TNF-α and offers a protective immune response to the host against *Candida* infections. Th1 has stimulatory effects on the phagocytosis and killing of *Candida* by neutrophils and macrophages and promotes optimal fungal clearance [16,32]. It has been demonstrated that mice deficient in the IFN-γ and IFN-γ receptor are highly susceptible to candidiasis, highlighting the importance of IFN-γ in host defense [33]. Both *C. albicans* and *C. glabrata* induce high levels of IFN-γ response and showed significant differences compared to *C. krusei*. This might be a difference in virulence factors between the species. The proinflammatory cytokines TNF-α, IL-1β, and IL-6 levels were increased upon *Candida* spp. stimulation compared to unstimulated and were higher during *C. albicans* infection. TNF-α showed increased levels in *Candida*-stimulated samples. There was a significant difference between *C. albicans* and *C. glabrata*, *C. albicans* and *C. tropicalis*, and *C. krusei* and *C. tropicalis*. Therefore, *C. albicans* induces strong IFN-γ and TNF-α compared to other species. IL-6 and IL-1β, both inflammatory cytokines, play a crucial role in inflammation in *Candida* infections. However, in our study, no significant differences in IL-6 and IL-1β levels were observed among *Candida* spp. Despite NAC species being less virulent than *C. albicans*, we observed similar IL-6 and IL-1β profiles for all the stimulations [34]. Similar to our study, Toth et al. reported that both TNF-α and IL-6 did not show a significant difference between *C. albicans* and *C. parapsilosis* [30]. The increased levels of these proinflammatory cytokines are crucial for driving the acute-phase immune response to fungal pathogens [26]. IL-12, predominantly produced by APCs, plays a crucial role in the differentiation of naive CD4 T-cells into Th1 subset [35]. In our investigation, the median levels of IL-12p70 were significantly higher in *C. albicans*- and *C. parapsilosis*-stimulated samples than *C. tropicalis* and *C. krusei*. This would be one of the major factors for the generation of Th1-type cytokine response (IFN-γ) against invasive candidiasis. This indicates an important role of IL-12/Th1 responses during *Candida* infection and this balance is needed for controlling systemic infections.

The Th2 immune response has been associated with increased fungal burden and disease exacerbation [10]. The inhibition of IL-4 signaling, and IL-10 knockout mice have been shown to enhance resistance to candidiasis [36]. In our study, the level of IL-4 was significantly lower compared to IL-10 levels. Furthermore, the baseline levels of anti-inflammatory cytokine IL-10 were notably lower than those of the proinflammatory cytokines. *C. glabrata* induced higher levels of IL-10 secretion than any other *Candida* spp. Therefore, our results presumably indicate that while *C. albicans* induces IFN-γ response, *C. glabrata* promotes IL-10 cellular responses. However, additional studies are required to confirm this statement. Though Th2 response has been considered nonprotective, they are required for the maintenance of a balanced non-deleterious proinflammatory Th1/Th17 response [37]. Several studies have reported that the production of Th1 cytokines is critical for controlling *Candida* infection, whereas Th2 cytokines counter-regulate the functions of Th1 cytokines [38,39]. In this line, it has been demonstrated in murine studies that overexpression of Th2 transcription factor (GATA-3) enhances susceptibility to systemic *Candida* infection, possibly by reducing the production of IFN-γ [10]. Thus, the paradigm between Th1 and Th2 cytokines is one of the important factors that will influence the outcome of infection.

We measured secreted IL-17A and IL-23 effector cytokines in the Th17 family, which induces the recruitment and activation of neutrophil granulocytes [40,41,42]. Recently, it has been demonstrated that *C. albicans*-specific T-cell response modulates antifungal Th17 response by cross-reactivity with other fungal species [43]. The cytokine IL-17A did not show a significant difference between the stimulations. Interestingly, the cytokine IL-23 showed a significant difference for *C. glabrata* compared to *C. parapsilosis* and *C. krusei*. Across all *Candida* spp. stimulations, the levels of secreted IL-17A and IL-23 were consistently lower in PBMCs. Interestingly, the median levels of IL-17A and IL-23 were slightly lower in *C. albicans* compared to *C. glabrata* stimulation. This could be due to the small frequency of IL-17-producing cells in the CD4/CD8 population, which was further confirmed by intracellular staining and flow cytometry. The percentage of IFN-γ and IL-17 response is not consistent with Toth et al.’s study, which reported significant differences in CD4^+^ IFN-γ^+^ and IL-17^+^ between *C. albicans* and *C. parapsilosis* [30]. This discrepancy might be due to the variation in the stimulation period and strain virulence nature. Though the levels of secreted IL-1β and IL-6 are substantial in all the stimulations, none showed a significant difference between *Candida* spp. In fact, both IL-1β and IL-6 are required for the initial differentiation of Th17 cells [44]. However, it does not account for the different Th17 responses evoked by *Candida* spp. Furthermore, Th17 response is involved in protection against mucosal infection and chronic mucocutaneous candidiasis, while Th1 response is predominant in invasive infections [12,29,42,45]. The differential immune response induced by distinct *Candida* spp. at the priming and effector stage is largely due to the different cytokine environments induced by each pathogen [46]. In summary, our study contributes to the better understanding of the immune response against *Candida* spp.

Some of the major limitations of our study include the lack of a candidiasis patient group and the small sample size. As a preliminary study, our primary goal was to analyze whether the selected *Candida* spp. shows a differential immune response to each species or not. However, to enrich our understanding of the spectrum of immune responses during *Candida* infection and to assess the *Candida* spp.-specific immune responses to discriminate healthy controls from the diseased group, including candidiasis patients for the ex vivo analysis would be beneficial. In addition, relevant differences between reference strains need to be confirmed with a broader collection of clinical isolates to support the claim that the detected differences are species-specific. However, we plan to include more clinical strains from each species and validate these findings in a larger sample in future studies. Another limitation is using heat-killed *Candida* cells, which may alter cell wall architecture and influence immune response differently compared to UV-killed cells. However, heat-killed *Candida* enhances immune response due to heat-induced permeabilization of the cell wall and subsequent exposure of the underlying β-1,3-glucan layer, which is strongly immunogenic [31,47]. Further, heat-killed *C. albicans* has been frequently used in immunological studies to prevent hyphae generation in response to serum components of cell culture media [30,31,48]. Additionally, we did not assess the molecular mechanisms that control this species-specific cytokine response. Although the underlying molecular mechanisms of the induction of these cytokines remain to be clarified, these findings may partly explain the differential defense patterns of these *Candida* spp. It would be important to study other innate, polyfunctional T-cell responses and conduct non-conventional T immunoprofiling to help in our overall understanding of host immune response. The current manuscript makes a useful addition and help us to understand the *Candida* species-specific T-cell response and subsets of *Candida* at the species level. Additional studies are highly recommended to validate our observation in larger study populations. Furthermore, we are interested in including the most important emerging clinical isolates such as *C. auris* and focusing on molecular mechanisms to better understand the host defense mechanisms.

## Figures and Tables

**Figure 1 biomedicines-12-01487-f001:**
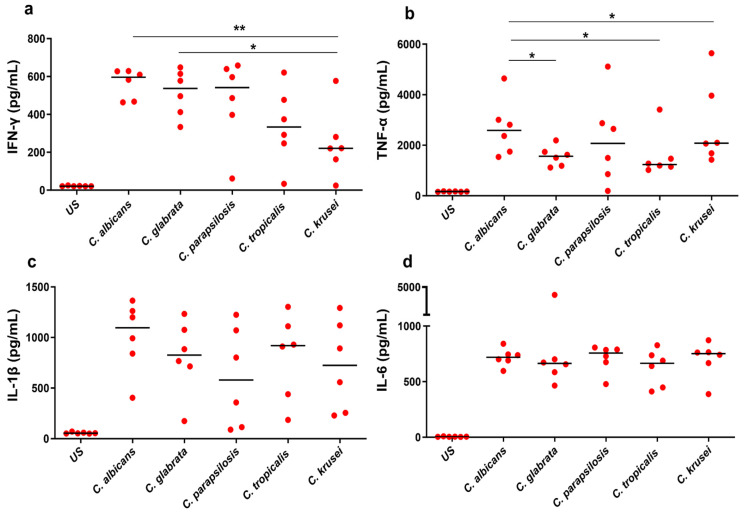
Proinflammatory cytokine response in response to *Candida* species: PBMCs from healthy controls were stimulated with *C. albicans*, *C. glabrata*, *C. parapsilosis*, *C. tropicalis,* and *C. krusei* for 24 h. Cells cultured under similar conditions without any stimulation served as an unstimulated (US) control. Supernatants were collected and the concentrations of all cytokines were measured by ELISA. (**a**) IFN-γ, (**b**) TNF-α, (**c**) IL-1β, (**d**) IL-6. Nonparametric Kruskal–Wallis ANOVA coupled with Dunn’s correction was used to compare between stimulations. Horizontal line indicates median. *p* value < 0.05 was considered statistically significant. * *p* < 0.05, ** *p* < 0.01.

**Figure 2 biomedicines-12-01487-f002:**
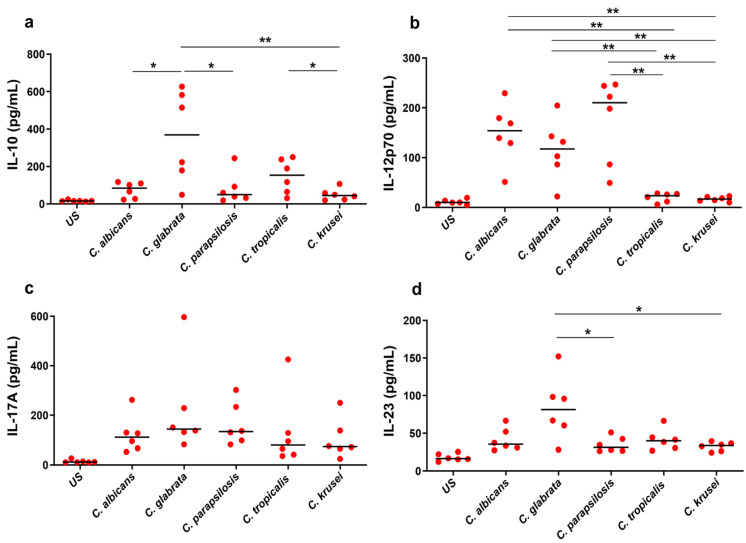
Anti-inflammatory, Th17, IL-12p40, and IL-23 cytokine response to *Candida* species: PBMCs from healthy controls were stimulated with *C. albicans*, *C. glabrata*, *C. parapsilosis*, *C. tropicalis,* and *C. krusei* for 24 h (IL-10, IL-12p40) or 6 days (IL-17A, IL-23). Cells cultured under similar conditions without any stimulation served as an unstimulated (US) control. Supernatants were collected and the concentration of all cytokines was measured by ELISA. (**a**) IL-10, (**b**) IL-12p40, (**c**) IL-17A, (**d**) IL-23. Nonparametric Kruskal–Wallis ANOVA coupled with Dunn’s correction was used to compare between multiple stimulations. Horizontal line indicates median. *p* value < 0.05 was considered as statistically significant. * *p* < 0.05, ** *p* < 0.01.

**Figure 3 biomedicines-12-01487-f003:**
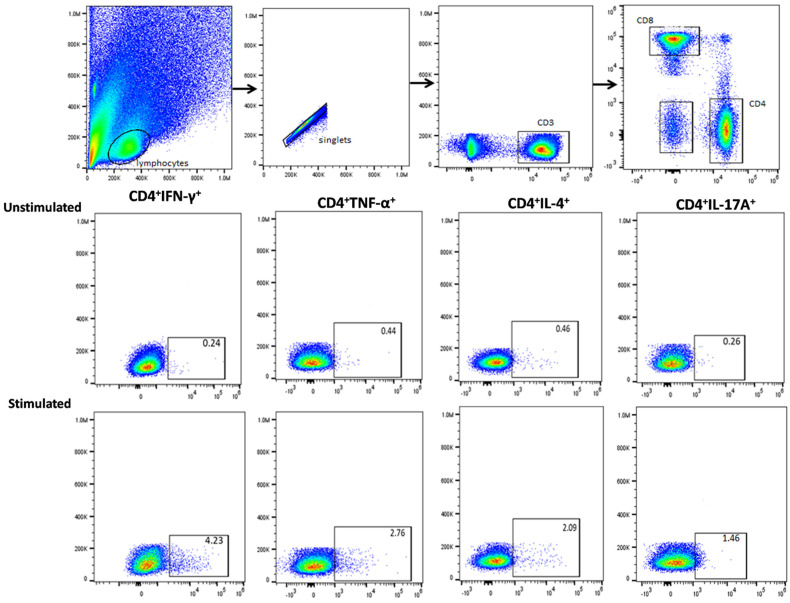
General gating strategy followed for CD4^+^ and CD8^+^ T-cells: Lymphocytes were gated based on size and granularity. From singlet cells, T lymphocytes were selected by the CD3^+^ marker and then CD4^+^ and CD8^+^ populations were selected., e.g., gating strategy followed for CD4 secreting IFN-γ, TNF-α, IL-4 and IL-17A in both unstimulated control and *C. albicans*-stimulated sample. Boundaries were defined by setting fluorescence-minus-one (FMO) controls.

**Figure 4 biomedicines-12-01487-f004:**
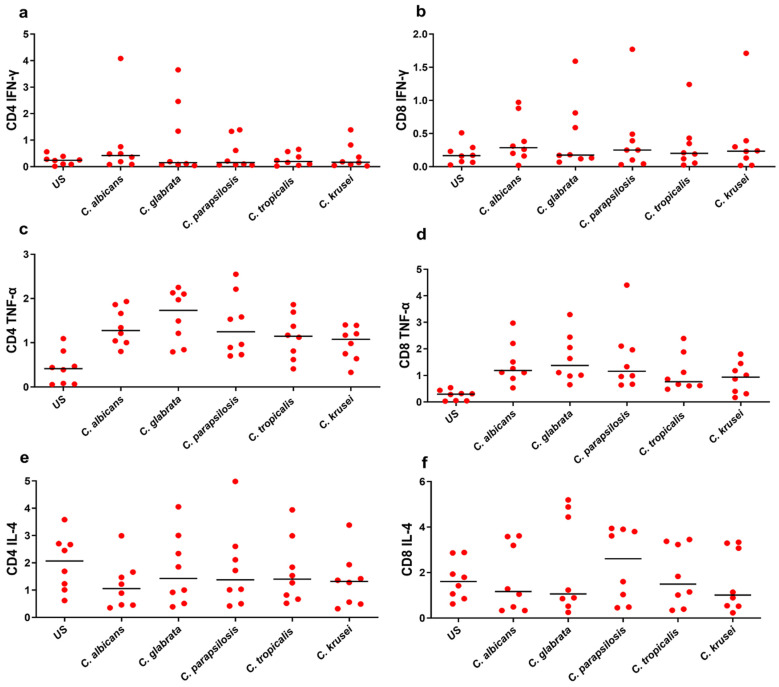
The scatter plot represents frequency of CD4^+^ and CD8^+^ secreting IFN-γ (**a**,**b**), TNF-α (**c**,**d**), and IL-4 (**e**,**f**) upon stimulating healthy human PBMCs with *Candida* spp., and no stimulus was added to unstimulated (US) control. Nonparametric Kruskal–Wallis ANOVA coupled with Dunn’s correction was used to compare between multiple stimulations. Horizontal line indicates median. Differences were compared using a Mann–Whitney U test. *p* value < 0.05 was considered statistically significant.

**Table 1 biomedicines-12-01487-t001:** The median levels of functional phenotypes of CD4 and CD8 T-cells against *Candida* spp. in 8 healthy controls.

Phenotype	US	*C. albicans*	*C. glabrata*	*C. parapsilosis*	*C. tropicalis*	*C. krusei*
CD4 IFN-γ	0.24	0.42	0.15	0.155	0.195	0.165
CD4 TNF-α	0.415	1.275	1.73	1.245	1.145	1.075
CD4 IL-4	2.07	1.055	1.425	1.375	1.4	1.32
CD4 IL-17A	0.235	0.4	0.645	0.51	0.455	0.505
CD4 IL-22	0.115	0.22	0.24	0.2	0.175	0.19
CD8 IFN-γ	0.165	0.285	0.175	0.25	0.2	0.235
CD8 TNF-α	0.295	1.19	1.375	1.16	0.765	0.94
CD8 IL-4	1.605	1.165	1.06	2.605	1.49	1.015
CD8 IL-17A	0.14	0.585	0.445	0.48	0.375	0.345
CD8 IL-22	0.1105	0.17	0.205	0.0565	0.17	0.185

## Data Availability

The data that support the findings of this study are available on request from the corresponding author. The data are not publicly available due to privacy or ethical restrictions.

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
