# Peer review of "Comparative Evaluation of Candida Species-Specific T-Cell Immune Response in Human Peripheral Blood Mononuclear Cells"

_biomedicines, 2024, doi:10.3390/biomedicines12071487_

Round 1

Reviewer 1 Report

Comments and Suggestions for Authors

The authors of the manuscript titled "Comparative evaluation of Candida species-specific T cell immune response in human peripheral blood mononuclear cells" compared the ability of cytokine production by human peripheral blood mononuclear cells upon stimulation with different candida species. The body of work presented here is appropriate for Biomedicines; however, it needs major revisions before it can be considered for publication.

Points that need to be addressed.

  1. The entire study was based on the stimulation of PBMCs from healthy donors with fungal species and comparing cytokine production. How significant would the difference be in cytokine production if the samples were from patients with proven invasive Candida infection? This comparison would provide a better understanding of immune response in both sick and healthy patients.
  2. The sample size is too small for this study. Do you think the age and gender of the blood donors have any effect on the immune response?
  3. In both figures 1 and 2, do the data points correspond to samples from 6 patients, or are they from 8 patients?
  4. Authors should redo both panels A and B of Figure 3. Please ensure that the figures have better clarity and are large so the readers can follow the data. Understanding the results in section 3.3 is challenging without better figures.
Comments on the Quality of English Language

Minor editing required

Author Response

Reviewer #1:

The authors of the manuscript titled "Comparative evaluation of Candida species-specific T cell immune response in human peripheral blood mononuclear cells" compared the ability of cytokine production by human peripheral blood mononuclear cells upon stimulation with different candida species. The body of work presented here is appropriate for Biomedicines; however, it needs major revisions before it can be considered for publication.

Points that need to be addressed.

  1. The entire study was based on the stimulation of PBMCs from healthy donors with fungal species and comparing cytokine production. How significant would the difference be in cytokine production if the samples were from patients with proven invasive Candida infection? This comparison would provide a better understanding of immune response in both sick and healthy patients.

We completely agree with the reviewer comment. This is one of the major limitations of our study. However, we plan to recruit invasive candidemia patient group and compare with healthy controls to evaluate the discriminative potential of these adaptive immune datasets. See the line numbers 383-388.

  1. The sample size is too small for this study. Do you think the age and gender of the blood donors have any effect on the immune response?

Thank you for the reviewer’s constructive comment. No statistical methods were used to predetermine the sample size. Since it is a pilot study, our primary goal is to analyze, whether the selected Candida species shows a differential immune response to each species or not. However, to enrich our understanding of the spectrum of immune responses during Candida infection, we plan to include more healthy controls and invasive candidemia patients to validate these findings.

Age and gender are one of the factors to influence the host immune response in invasive candidiasis infection. It has been reported that the extreme of age is associated with development of invasive candidiasis infections (Laupland KB et al., 2005, Invasive Candida species infections: a 5-year population-based assessment). Similarly, meta-analyses/systematic reviews revealed that males represented from 52.5% to 64.7% of invasive candidiasis cases (Matthias et al., 2022, Let's talk about sex characteristics—As a risk factor for invasive fungal diseases).

In the current study, due to limited number of study subjects, we didn’t check the subgroup analysis based on age and gender. There is a possibility to incur type II error in this small subgroup comparison.

  1. In both figures 1 and 2, do the data points correspond to samples from 6 patients, or are they from 8 patients?

Thanks for the opportunity to clarify this discrepancy. We have recruited 8 healthy controls, and we generated the flow cytometry immunoprofiling data for all the healthy control samples. As we mentioned in the methods, we have performed the ELISA in 6 healthy control samples due to insufficient amount of blood. See the line numbers 168-169.

  1. Authors should redo both panels A and B of Figure 3. Please ensure that the figures have better clarity and are large so the readers can follow the data. Understanding the results in section 3.3 is challenging without better figures.

As per the reviewer comment, Figure 3 has been reframed. The resolution of figure 3 has been improved.

Reviewer 2 Report

Comments and Suggestions for Authors

in this manuscript, Pathakumari et al, ,present their finding from the study of interaction between  PBMCs and several Candida spp (heat-killed).  The immune response is mainly assessed  by terms of cytokine production by T cells subpopulations.. 

The subject of the study is very interesting and the manuscript is overall well-written.

The main drawback is the fact that each species is represented by only one isolate. In my opinion, we could extract a conclusion for the immune response to a Candida species if we studied a number of (clinical isolates). 

So, I could characterize the results as preliminary. Nevertheless, if the authors cannot extent the number of isolates studied, then, they should underline that further experiments should be performed. 

minor points

line 17, 40:candidemia

line 42 rewrite as"infections cases caused by Candida albicans (C. albicans), ...."

line 48: from. ow on use only "C. albicans"

line64-67 please rewrite

line 68 "it"

line 84-85: this cannot be accurate. if you refer to a specific study please specify this

line 96 "C. albicans"

Comments on the Quality of English Language

The use of English language is fine

Author Response

Reviewer #2:

Comments and Suggestions for Authors

in this manuscript, Pathakumari et al, present their finding from the study of interaction between PBMCs and several Candida spp (heat-killed).  The immune response is mainly assessed by terms of cytokine production by T cells subpopulations. The subject of the study is very interesting, and the manuscript is overall well-written. The main drawback is the fact that each species is represented by only one isolate. In my opinion, we could extract a conclusion for the immune response to a Candida species if we studied a number of (clinical isolates). So, I could characterize the results as preliminary. Nevertheless, if the authors cannot extent the number of isolates studied, then, they should underline that further experiments should be performed.

Thank you for the reviewer’s constructive comment. Since it is a pilot study, our primary goal is to analyze, whether the selected Candida species shows a differential immune response to each species or not. However, to enrich our understanding of the spectrum of immune responses during Candida infection, we plan to include more clinical strains from each species and validate these findings in more sample size in future studies.

We have added this limitation in the text in line numbers 389-392.

minor points

line 17, 40: candidemia

Replaced Candidemia with small letter "c" and maintained consistently throughout the manuscript.

line 42 rewrite as"infections cases caused by Candida albicans (C. albicans), ...."

As per reviewer comment, this sentence has been changed. See line number 41 in revised version.

line 48: from now on use only "C. albicans"

As per reviewer comment, we used C. albicans throughout the manuscript.

Line 64-67 please rewrite

As per reviewer comment we reframed this sentence as per the reference [12]. Line numbers 64-66. In revised version

line 68 "it"

Thank you for identifying these typos. The word “It” has been changed to “it”.

line 84-85: this cannot be accurate. if you refer to a specific study please specify this.

We apologize for this mistake. We have rewritten this sentence as per the reference. See the line numbers 83-84 in revised manuscript.

line 96 "C. albicans"

Thank you for identifying these errors. We have changed the c. albicans to “C. albicans” and maintained consistently throughout the manuscript.

Round 2

Reviewer 1 Report

Comments and Suggestions for Authors

Addressed the comments

Reviewer 2 Report

Comments and Suggestions for Authors

Dear authors,

in my opinion the presentation of the manuscript has been substantially improved and it can be eligible for publication.

minor points

line 65: "oro..." not capital "Oro.."

line 83: "bloodstream" instead of "blood stream"

Comments on the Quality of English Language

in my opinion, the use of English language is fine.